# Combining Essential Oils with Each Other and with Clotrimazole Prevents the Formation of *Candida* Biofilms and Eradicates Mature Biofilms

**DOI:** 10.3390/pharmaceutics14091872

**Published:** 2022-09-05

**Authors:** Rafael Alves da Silva, Nagela Bernadelli Sousa Silva, Carlos Henrique Gomes Martins, Regina Helena Pires, Denise Von Dolinger de Brito Röder, Reginaldo dos Santos Pedroso

**Affiliations:** 1Post-Graduation Program in Health Sciences, Federal University of Uberlândia (UFU), Uberlândia 38400-902, MG, Brazil; 2Post-Graduation Program in Applied Immunology and Parasitology (PIPA), Federal University of Uberlândia (UFU), Uberlândia 38400-902, MG, Brazil; 3Institute of Biomedical Sciences (ICBIM), Federal University of Uberlândia (UFU), Uberlândia 38400-902, MG, Brazil; 4Post-Graduation Program in Health Promotion, University of Franca (UNIFRAN), Franca 14404-600, SP, Brazil; 5Technical School of Health (ESTES), Federal University of Uberlândia (UFU), Uberlândia 38400-902, MG, Brazil

**Keywords:** essential oils, clotrimazole, *Candida* spp., synergy, biofilm, toxicity, *Caenorhabditis elegans*

## Abstract

Fungal infections by *Candida* spp. are opportunistic and most often occur in individuals with some predisposing factor. Essential oils (EO) have anti-*Candida* potential, being a therapeutic alternative to be explored, especially for superficial and mucosal candidiasis. The objective was to analyze the synergistic potential between the EO of *Citrus limon*, *Cupressus sempervirens*, *Litsea cubeba* and *Melaleuca alternifolia*, and each of them with clotrimazole, to inhibit in vitro the formation and eradication of *Candida* spp. biofilms. Added to this, the survival of *Caenorhabditis elegans* was evaluated after exposure to EO, clotrimazole and their synergistic combinations. Anti-*Candida* activity was determined by microdilution for the substances alone and in EO–EO and EO–clotrimazole combinations. The combinations were performed by the checkerboard method, and the reduction in the metabolic activity of biofilms was determined by the viability of MTT/menadione. *C. elegans* larvae survival was evaluated after 24 h of exposure to EO, clotrimazole and synergistic combinations. The minimum inhibitory concentration (MIC) of EO ranged from 500 to >4000 µg/mL. The lowest MIC (500 µg/mL) was for *C. sempervirens* and *L. cubeba* on a *C. krusei* isolate; for clotrimazole, the MIC ranged from 0.015 to 0.5 µg/mL. Biofilm inhibition and eradication both ranged from 1000 to >4000 µg/mL. The lethal concentration (LC_50_) of *C. limon*, *L. cubeba* and *M. alternifolia* was 2000 µg/mL for *C. elegans*, while for *C. sempervirens* and clotrimazole, it was not determined within the concentration limits tested. In combination, more than 85% of the larvae survived *M. alternifolia*–clotrimazole, *M. alternifolia*–*L. cubeba*, *C. sempervirens*–clotrimazole and *C. sempervirens*–*C. limon* combinations. This study is the first, to our knowledge, to present a synergistic relationship of EO–EO and EO–clotrimazole combinations on *Candida* spp. biofilms.

## 1. Introduction

Infections caused by species of the genus *Candida* are opportunistic and more severe in immunocompromised, hospitalized individuals, using invasive devices and with comorbidities [1]. Superficial candidiasis affects the oral and vaginal mucosa, skin and nails; and factors external to the individual, such as climatic conditions in tropical and subtropical regions, and factors inherent to it, such as local humidity, use of immunosuppressive or antibacterial drugs and some comorbidities, such as diabetes, facilitate the development of the disease [2]. In superficial infections, the most frequent *Candida* species is *C. albicans*; however, in recent years, *Candida* non-*albicans* species have shown relevance among the causative agents of vulvovaginal candidiasis (VVC) and recurrent VVC (RVVC), including the species *C. glabrata*, *C. krusei* and *C. tropicalis* [2,3,4].

The azole antifungals have been one of the therapeutic options for the treatment of superficial mycoses since the 1960s–1970s [5,6]. In this sense, topical formulations containing azoles are attractive for VVC and RVVC due to the lower incidence of adverse effects compared to the same drug class for oral use and systemic action [3]. Clotrimazole has a cure rate of between 73% and 100% of infections, similar to other topical antifungals such as nystatin [6,7].

In the last decades, the report of new *Candida* species and in vitro resistant isolates to traditional antifungals has been an incentive for the search and development of new ways of managing these infections [8]. Resistance is a result of multiple factors that include structural changes in the drug target and the ability of *Candida* spp. to form biofilms [9,10,11]. In this sense, the community structure and firm adherence between the microorganisms of the biofilm allow a barrier condition that makes penetration of drugs difficult and consequently reduces the effectiveness of the treatment [8,10,11].

Essential oils are plant-derived products with potential activity against microorganisms, attributable to the complex mixture of chemotypes [12,13,14]. Recently, combinations of multiple agents have optimized antifungal activity against clinically relevant fungi. Thus, a new therapeutic approach combining conventional antifungal drugs, such as clotrimazole, and natural products with antifungal activity may have the potential for clinical use [15,16,17,18].

The in vivo screening of compounds with proven in vitro antimicrobial action is one of the necessary steps within the current safety context to identify the toxicity of new anti-infective agents [19]. In this context, in vivo studies using alternative animal models such as *Drosophila melanogaster*, *Galleria mellonella* and *Caenorhabditis elegans* have been proposed to assess the preliminary toxicity of new health products [20,21]. Thus, the free-living nematode *C. elegans* can be an alternative predictive model option, being of low cost, fast cultivation and not very complex laboratory handling, and lending itself to the evaluation and screening of acute toxicity for use in animals, including humans, and contamination of the environment [11,19,22]. In this sense, the evident anti-*Candida* action of isolated essential oils could mean they present lower inhibitory values when combined with other essential oils or antifungal substances, such as clotrimazole, and its acute toxic repercussions.

Thus, in this study, the in vitro inhibitory activity of the essential oils of *Cupressus sempervirens*, *Citrus limon*, *Litsea cubeba* and *Melaleuca alternifolia*, alone and in combination, and associated with clotrimazole, against *Candida* species biofilms were analyzed. Furthermore, the in vivo toxicity of these essential oils against *C. elegans* was also evaluated.

## 2. Materials and Methods

### 2.1. Essential Oils and Candida Species

The essential oils (EO) of *Citrus limon*, *Cupressus sempervirens*, *Litsea cubeba* and *Melaleuca alternifolia* (FERQUIMA^®^; Vargem Grande Paulista, SP, Brazil) were included in this study. The analysis of the EO, carried out by chromatography, was informed by the supplier company and is shown in Table 1. Four reference strains, *Candida albicans* ATCC 90028, *Candida glabrata* ATCC 2001, *Candida krusei* ATCC 6258 and *Candida parapsilosis* ATCC 22019, and four clinical isolates from the vaginal mucosa (*Candida albicans* SV 01, *Candida glabrata* SV 02, *Candida krusei* SV 03 and *Candida parapsilosis* SV 04) obtained from previous studies were included in this study [23]. All microorganisms were stored in brain heart infusion (BHI)–glycerol broth at −20 °C and subcultured on Sabouraud dextrose agar (SDA; Disco, Detroit, MI, USA) and CHROMagar *Candida* medium (Becton Dickinson and Company, Sparks, MD, USA), to evaluate the viability and purity, and even to confirm the identification of the species.

### 2.2. Determination of the Minimum Inhibitory Concentration (MIC) and Minimum Fungicidal Concentration (MFC)

To determine the MIC of EO and antifungal agents, the broth microdilution method was used [24], with some adaptations. Flat-bottomed 96-well plates (Kasvi, PR, Brazil) and RPMI-1640 broth with glutamine and without sodium bicarbonate (Corning Incorporated, Corning, NY, USA) were used, plus 18 g/L of glucose (Sigma-Aldrich, St. Louis, MO, USA), buffered with MOPS at pH 7 (Sigma-Aldrich, St. Louis, MO, USA) as a culture medium, and yeast suspension at a resulting final concentration of 0.5–2.5 × 10^3^ CFU/mL. The concentration ranges varied from 7.81 to 4000 µg/mL for EO, from 0.03 to 16 µg/mL for amphotericin B (Sigma-Aldrich, St. Louis, MO, USA) and from 0.125 to 64 µg/mL for fluconazole and clotrimazole (Sigma-Aldrich, St. Louis, MO, USA). Amphotericin B and fluconazole were used as controls [24].

EO and amphotericin B were solubilized in DMSO (dimethyl sulfoxide; 2%), fluconazole and clotrimazole in water and later diluted in RPMI. *C. parapsilosis* ATCC 22019 and *C. krusei* ATCC 6258 strains were used to validate the tests [24]. The MIC was determined in a spectrophotometer at 490 nm [24]. The cut-off point for defining susceptibility was set at 80% inhibition of fungal growth compared to azole-free growth and 90% inhibition for amphotericin B [24] and EO [25]; cultures were incubated for 24 h at 35 °C. The tests were performed in triplicate.

The MFC was determined by transferring 5 μL of cell suspension from each well to a plate containing SDA, followed by incubation for 48 h at 30 °C. The MFC was the one corresponding to the concentration of the well where the growth of yeast colonies was no longer evident [26]. The ratio of the MFC and MIC of EO and clotrimazole was used to interpret the results, defining the drug as fungistatic (MFC/MIC: >4) or fungicidal (MFC/MIC: ≤4) [27].

### 2.3. Evaluation of the Activity of EO and Clotrimazole against Candida spp. Biofilms

#### 2.3.1. Determination of the Minimum Biofilm-Inhibiting Concentration (MBIC)

Inhibition of biofilm formation was determined in 96-well, flat-bottomed plates [28], to which 100 µL of cell suspension in RPMI-1640 was added (1 to 5 × 10^6^ cells/mL, adjusted to a turbidity equivalent to 0.5 McFarland scale), as well as 100 µL of the drug (EO and/or clotrimazole), at concentrations of 4 × MIC, 2 × MIC, 1 × MIC, 0.5 × MIC and 0.25 × MIC. The culture was incubated at 35 °C for 48 h. Then, non-adherent cells were removed, and the wells were washed three times with PBS (10 mM phosphate buffer, 2.7 mM potassium chloride, 137 mM sodium chloride, pH 7.4). Then, 100 µL of MTT solution (5 mg/mL; 3-(4,5-dimethyl-2-thiazolyl)-2,5-diphenyl-2H tetrazolium bromide; Sigma-Aldrich, St. Louis, MO, USA) plus 10 µL of phytomenadione (1 µM; 2-methyl-3-[(E,7R,11R)-3,7,11,15-tetramethylhexadec-2-enyl]naphthalene-1, 4-dione; Sigma-Aldrich, St. Louis, MO, USA) was added to each well. The plate was incubated at 35 °C for 24 h in the dark. Subsequently, the supernatant was removed, and 100 µL of DMSO was added to each well, and the plate was incubated at 35 °C for 15 min, protected from light. Then, 80 µL of solvent was removed from each well and transferred to another plate, and the reading was performed at 490 nm [29]. Growth and sterility controls were included for each plate in the experiment. The tests were performed in triplicate.

#### 2.3.2. Determination of the Minimum Biofilm-Eradication Concentration (MBEC)

The biofilm was previously formed in 96-well, flat-bottomed plates. Two hundred microliters of the yeast cell suspension were added to each well (1 to 5 × 10^6^ cells/mL, adjusted to 0.5 McFarland in RPMI-1640), and the plates were incubated at 35 °C for 48 h [28]. Then, non-adherent cells were removed, and the wells were washed three times with PBS. Then, 100 µL of the drug (EO and/or clotrimazole) was added at concentrations of 4 × MIC, 2 × MIC, 1 × MIC, 0.5 × MIC and 0.25 × MIC and incubated at 35 °C for 24 h in the dark. The procedures for revealing the biofilm were the same as described in the previous item.

### 2.4. Evaluation of the Synergistic Potential of EO–EO and EO–Clotrimazole Associations against Planktonic Growth and on Biofilms

To evaluate the combined effect of EO and clotrimazole on planktonic cells, the checkerboard technique was used [30,31]. The synergistic potential of the combination of the EO of *C. sempervirens*, *C. limon*, *L. cubeba* and *M. alternifolia* among themselves and of each one of them with clotrimazole on the *Candida* biofilm was made according to the results obtained for the MIC of the planktonic cells, as provided in the Appendix A. Concentrations of 0.25 × MIC, 0.5 × MIC, 1 × MIC, 2 × MIC and 4 × MIC were used for drug testing. Then, 100 µL of drug A (EO) horizontal and 100 μL of drug B (EO/clotrimazole) vertical, and 100 μL of cell suspension (1 to 5 × 10^3^ cells/mL) were added to all wells of a 96-well flat-bottomed plate. Growth (containing drug-free yeast suspension) and sterility (RPMI-1640) controls were included in each plate. The plates were incubated at 35 °C for 48 h, and the reading was performed at 490 nm; results were considered capable of reducing ≥ 90% of optical density (OD) in relation to the control free of EO and clotrimazole [30]. The results were interpreted according to the fractional inhibitory concentration index (FICI), determined as follows:(1)$$\mathrm{FICI}:\;\left(\frac{\mathrm{MIC}\;“A”\;\mathrm{combined}}{\mathrm{MIC}\;“A”\;\mathrm{isolated}}\right)+\left(\frac{\mathrm{MIC}\;“B”\;\mathrm{combined}}{\mathrm{MIC}\;“B”\;\mathrm{isolated}}\right)$$

The interpretation was conducted according to the classification of the substance interaction score, where antagonism was considered when the score was greater than 4.0, indifference at a score greater than 1, additivity at a score between 0.5 and 1.0, and synergism at a score less than 0.5 [31]. One hundred microliters of the combined solution “A” (EO) and “B” (EO or clotrimazole) was added to each well, starting from column 11 and column 2. Briefly, the first and last wells received the highest concentrations of each of the two compounds evaluated, resulting in decreasing concentrations from one end of the plate to the other. Growth and sterility controls were included in each of the plates, and each experiment was conducted in triplicate.

### 2.5. Toxicity Assay for Caenorhabditis elegans

The toxicity test was performed by exposing *C. elegans* larvae (AU37 [glp-4 (bn2) I; mutant strain sek-1 (km4) X) to EO and clotrimazole [25].

*C. elegans* larvae were transferred to nematode growth medium (NGM), contained in Petri dishes, which contained a previous mat of *Escherichia coli* OP50 (*E. coli*). The plates were incubated at 16 °C for 72 h. Then, synchronization of the larvae in stage L2 was performed by treating the larvae with sodium hypochlorite. Then, the larvae were transferred to another plate containing NGM medium without *E. coli* OP50 and incubated at 16 °C for 24 h [20,32].

For the experiment, a solution medium, composed of 40% BHI broth, plus cholesterol (10 µg/mL), kanamycin (90 µg/mL), ampicillin (200 µg/mL) and 60% 50 mM NaCl, was used. The assay was performed using 96-well flat-bottomed plates. Then, 180 µL of solution medium and 20 µL of the suspension of synchronized larvae in stage L4 were added to each well of the plate so that 10 to 20 *C. elegans* larvae were placed in each well, evaluated in final serial concentrations ranging from 4000 to 250 µg/mL diluted in solution medium. As a survival control, solution medium plus the larvae, without drug, was used, and as a test control, solution medium and DMSO were used. The plates were incubated for 24 h at 35 °C in a humid chamber.

The results were interpreted considering the survival rate of the larvae and the 50% lethal dose (LD_50_), determined by the concentration of the drug that was able to kill 50% of the larvae [33,34]. Each experiment was performed twice in triplicate.

## 3. Results

### 3.1. Determination of the MIC and MFC of Essential and Antifungal Oils against Planktonic Growth

The lowest MIC (500 µg/mL) found was for *C. krusei* SV 03, with the EO of *C. sempervirens* and *L. cubeba*. The MIC of the EO ranged from 500 to >4000 µg/mL, considering the different oils and the eight isolates tested. The EO of *C. limon* presented an MIC that ranged from 1000 to 4000 µg/mL, *C. sempervirens* from 500 to >4000 µg/mL, *L. cubeba* from 500 to 2000 µg/mL and *M. alternifolia* from 1000 to 2000 µg/mL. For clotrimazole, the MIC ranged from 0.015 to 0.5 µg/mL (Table 1). The MIC of fluconazole and amphotericin B (test validation controls) were, respectively, 32 and 1 µg/mL; those for *C. krusei* ATCC 6258 and *C. parapsilosis* ATCC 2019 were 0.25 and 0.25 µg/mL, respectively.

The lowest fungicidal concentrations (1000 µg/mL) were found for the EO of *C. limon* against *C. albicans* ATCC 90028, *C. sempervirens* against *C. krusei* SV 03 and *L. cubeba* against *C. albicans* ATCC 90028 and *C. krusei* SV 03; for clotrimazole, the lowest fungicidal concentration (0.030 µg/mL) was found for the isolate of *C. glabrata* ATCC 2001.

Evaluation of the fungicidal activity (MFC/MIC: ≤4) showed that the EO of *L. cubeba* and the antifungal clotrimazole were fungicidal for all the tested isolates; however, all the other EO evaluated presented fungicidal activity dependent on the isolate, as can be seen in Table 2. Thus, fungicidal activity was found for the EO of *C. limon*, *M. alternifolia* and *C. sempervirens*, respectively, for four, five and six isolates.

### 3.2. Assessment of the Development of Candida spp. Biofilms

The activity of EO and clotrimazole to inhibit (MBIC) and eradicate (MBEC) the biofilm formed by *Candida* species is shown in Table 3. Most EO presented an MBIC greater than or equal to 4000 µg/mL. The lowest MBIC was 1000 µg/mL, found for *C. sempervirens* (*C. krusei* SV 03), *L. cubeba* (*C. albicans* ATCC 90028) and *M. alternifolia* (*C. krusei* SV 03). The lowest MBEC was 1000 µg/mL for *L. cubeba* (*C. albicans* ATCC 90028). Clotrimazole demonstrated MBIC and MBEC values ranging from 0.125 to 2 µg/mL and 0.25 to 4 µg/mL, respectively.

### 3.3. Evaluation of Synergism of EO and Clotrimazole

The tests of EO–EO and EO–clotrimazole combinations resulted in 80 combinations; of these, 13 (16.25%) showed antagonism, 42 (52.5%) showed indifference, 17 (21.25%) had an additive effect and 8 (10%) showed synergism. The OE–OE and OE–clotrimazole combinations performed and their results related to inhibition of *Candida* spp. are provided in the Appendix A. The synergistic effect was variable, depending on the combination (EO–EO or EO–clotrimazole) and the *Candida* strain. The EO–EO and EO–clotrimazole combinations that showed synergism in the evaluation of MIC were selected for evaluation of inhibition (MBIC) and eradication (MBEC) of biofilms (Table 4).

### 3.4. In Vivo Assay in Caenorhabditis elegans

The test of acute toxicity of EO and clotrimazole against *C. elegans* showed average survival greater than 90% of larvae for the concentration of 250 µg/mL of all tested EO; and for clotrimazole, 100% of *C. elegans* larvae survived at all concentrations within the evaluated range (0.125–4 µg/mL). It was not possible to determine the LC_50_ of *C. sempervirens* because at all concentrations evaluated, survival occurred in more than 90% of the larvae; the LC_50_ for the EO of *M. alternifolia* was 2000 µg/mL and for *C. limon* and *L. cubeba* it was 4000 µg/mL (Figure 1; Table 5). The test controls showed that DMSO concentrations ≤5% did not affect the survival of *C. elegans* larvae, and the untreated control showed a mean survival of 96% at 24 h.

The EO–EO and EO–clotrimazole combinations that showed synergism in the evaluation of the MIC were selected to evaluate the survival of *C. elegans* larvae. Overall, the combinations showed a mean survival of 90% of the larvae for the combinations *L. cubeba*–*M. alternifolia*, *C. sempervirens*–*C. limon*, *M. alternifolia*–clotrimazole and *C. sempervirens*–clotrimazole (Table 6). Survival of less than 20% of larvae, demonstrating greater acute toxicity, was found for the combinations *C. limon*–*M. alternifolia* and *L. cubeba*–*C. limon*.

## 4. Discussion

EO have been extensively studied nowadays and can be a complementary alternative for the treatment of infections caused by *Candida* species, especially mucocutaneous infections. This study investigated the activity of the EO of *C. limon*, *C. sempervirens*, *L. cubeba* and *M. alternifolia*, alone and in combination with each other and with clotrimazole, on four species of the genus *Candida*, to determine in vitro the MIC, MBIC and MBEC; in addition to this, an in vivo toxicity assessment for the nematode *C. elegans* was performed.

The EO extracted from plants of the studied species are products that have a complex chemical composition and may have more than 20 identified compounds [12,13,14,15,18]. Our study used EO that presented limonene (65.6%), α-pinene (52.4%), terpinen-4-ol (41%) and geranyl acetate (42%) as the main component, respectively, for *C. limon*, *C. sempervirens*, *M. alternifolia* and *L. cubeba*. These constituents are like those described for EO of these plants in other studies [15,18,29,35,36,37,38,39,40,41]. Terpene derivatives, a class that includes the mentioned constituents, are closely related to the antimicrobial biological action of these EO, as already demonstrated for *Candida* spp. in other studies [13,15,16,41,42,43].

In the present study, the MIC varied according to the isolate and according to the EO, but the EO of *C. sempervirens* and *L. cubeba* presented the lowest MIC (500 µg/mL) for the same species (*C. krusei* SV 03), while *M. alternifolia* and clotrimazole combined (62.5–0.25 µg/mL) inhibited *C. albicans* ATCC 90028 at lower concentrations than in isolation. In this sense, the ranges of results for the EO showed the effectiveness of plant-derived products in inhibiting microorganisms [42,43], a significant finding for the genus, given the recognized adaptive antifungal arsenal associated with *C.*
*albicans* and the intrinsic fluconazole resistance of *C. krusei* [9,20,44].

This variability of MIC can be observed in the literature [9,18] and is due to the characteristics of each isolate, which may be related to virulence factors and the origin of the isolate (blood, feces, respiratory tract or environment). In addition to this, storage and the constant activation and reactivation of cells, which occur in repeated cultures, may have generated adaptive changes in the phenotypic profile of the reference strains [45].

The MFC were, on average, 2 × MIC for most isolates and EO, but for some, it was not possible to make this determination, as the values were greater than 4000 µg/mL, that is, greater than the limits of concentrations tested. Still, MIC and MFC of 2000 and 1000 µg/mL, respectively, were observed for *C. albicans* ATCC 90028 when evaluated for the EO of *C. limon*. This fact may be explained if the growth curve of cells in contact with this EO is evaluated, as it is possible that the fungicidal effect occurs through mechanisms that involve the depletion of some essential intracellular constituent for growth, such as ergosterol reserves, associated with other mechanisms of enzymatic inhibition, or action on the membrane or cell wall [46] which is time-dependent, but other assays need to be performed to elucidate this finding.

The anti-*Candida* activity of EO may be a direct result of the interaction of the various chemical components present and the association of different mechanisms, which may explain the fungistatic and fungicidal effects. The characteristics common to EO, such as lipophilicity and ability to cause damage to vital structures, membrane, and cell wall, result in increased membrane permeability and release of intracellular contents, with consequent death of *Candida* spp. cells [13,37].

The fungistatic action of clotrimazole at low concentrations is due to structural changes in ergosterol; at high concentrations, it has a fungicidal effect [6]. Thus, it can be assumed that the potentiation of the effects, demonstrated by the synergism observed in the association of clotrimazole with different proportions of EO, is the result of the multiplicity of mechanisms resulting from the various constituents of the EO, leading to the fungicidal effect [47]. The activity of EO against biofilm [10,13,17,25,26,48,49,50,51] is another factor that contributes to the need for studies that evaluate the combination of other drugs and a greater number of EO [26,47,51,52].

The application of a product with simultaneous inhibition of microbial growth and biofilm is advantageous since it allows for more efficient satisfactory results in different structures of *Candida* spp. The present study demonstrated that there was inhibition of biofilm formation and a reduction in the viability of the cells of previously formed biofilm, with MIC up to five times lower for the synergistic combination when compared to the same MIC found for the drugs evaluated alone. This study is the first, to our knowledge, to present a synergistic relationship of EO–EO and EO–clotrimazole on *Candida* spp., evaluating their action on biofilms.

The initial assessment of a substance, such as toxicity and antifungal activity, is a preliminary step in the design of new drug and health product candidates [32]. Our study sought to evaluate the safety of EO and clotrimazole alone, as well as in combinations, exposed for 24 h to the in vivo model *C. elegans*. It was observed that more than 80% of *C. elegans* larvae survived at concentrations of 500 µg/mL for three of the evaluated EO. For *C. sempervirens*, 80% of the larvae survived at the concentration of 4000 µg/mL, and for clotrimazole, the survival of 100% of the larvae was observed at all concentrations.

Among the EO evaluated, the biological activity of the EO of *C. limon* and *M. alternifolia* is better known when compared with those of *L. cubeba* and *C. sempervirens* [18]. As observed in Table 4, the LC_50_ was not determined for the EO of *C. sempervirens* (LC_50_: >4000 µg/mL), suggesting that it is the least toxic for *C. elegans* larvae among the four evaluated. Our study demonstrated that lethal toxicity of *L. cubeba* EO against *C. elegans* larvae was at 2000 µg/mL; however, lower concentrations such as 0.120–0.525 mg/mL (120–525 µg/mL) were found previously for the nematode *Bursaphelenchus xylophilus* [37]. In our study, we found lower toxicity of *C. sempervirens* EO alone; however, it was moderate and high for other combinations (OE–OE and OE–clotrimazole). Some studies have provided other models for assessing toxicity by evaluating different cell cultures, showing that in vitro inhibitory concentrations (IC_50_) for MCF-7 and MDA-MB-231 mammary tumor cells were lower than 34.5 and 65.2 μg/mL, respectively [53], and that *C. sempervirens* is lethal at higher concentrations in human promyelocytic leukemia strains (HL-60 and NB4) (LC_50_: 333.79 to 365.41 µg/mL) [38] and in experimental animal Ehrlich ascitic carcinoma (LC_50_: 372.43 µg/mL) [38]. In the larvae of *Culex quinquefasciatus*, a non-vertebrate model and vector of filariasis, the LC_50_ was 16.1 μg/mL after 24 h of exposure [39].

The complexity of factors intrinsic to EO, such as the variability and concentration of chemotypes, which can vary in the same plant species according to the part of the plant used for extraction, region of cultivation and stage of development, may be, in part, responsible for the different results obtained in the same toxicity model used. In different models, this variability of constituents can be even greater, as can be seen in some studies [12,15,18,41,49,53]. Thus, it is suggested that toxicity is evaluated in different models to obtain evidence of greater safety and definition of the best drug concentrations that may have biological action and an absence or reduction of damage.

Our study focused on the preliminary assessment of EO–EO and EO–clotrimazole combinations, using concentration ranges applied predictively to planktonic cells and subsequently to biofilm and *C. elegans* after 24 h. Therefore, the totality of combinations that the checkerboard provides for the biofilm was not explored, nor was the influence of different exposure times of the substances for inhibition, eradication and toxicity. Our study used evaluation in the *C. elegans* model; therefore, it is important to evaluate correlation with the results in other models for a better understanding of the mechanism related to toxicity, including the use of EO in biocompatible pharmaceutical applications in nanosystems to improve aspects of physicochemical and biological agents against *Candida* spp.

The complexity of the composition of EO allows wide use in alternative and complementary medicine. The exploration of antimicrobial activity may enable new strategies and therapeutic alternatives for infectious diseases, especially mucocutaneous ones, where topical application is possible. The association of EO makes it possible for some constituents, even though they are not in the majority, to interact, enhancing or evidencing biological effects and reducing toxicity. In this context, studies still need to be carried out to determine the practical relevance of the combinations, better concentrations of each one of them, and the economic and market viability, in addition to the advantages over existing products.

## 5. Conclusions

The EO–EO and EO–clotrimazole combinations showed synergistic activity in vitro, dependent on the isolate and on the *Candida* species, and of the combined drugs, when evaluating the inhibition of planktonic growth in vitro and the inhibition of biofilm formation and eradication. The combinations *M. alternifolia*–clotrimazole, *L. cubeba*–*M. alternifolia*, *C. sempervirens*–clotrimazole and *C. sempervirens*–*C. limon* were the most efficient against planktonic cells and biofilm. In addition, they demonstrated low or negligible toxicity to *C. elegans* larvae. Thus, our results suggest that the drug combinations evaluated here show promising activity in the control and treatment of vaginal infections caused by *Candida* species, for topical application through different devices, for example, local nanorelease systems, such as mucoadhesive formulations.

## Figures and Tables

**Figure 1 pharmaceutics-14-01872-f001:**
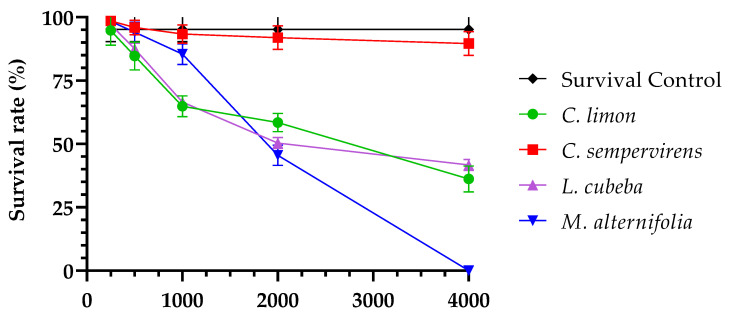
Survival rate (%) of *C. elegans* larvae in 24 h, evaluated at different concentrations of essential oils of *C. limon*, *C. sempervirens*, *L. cubeba* and *M. alternifolia*.

**Table 1 pharmaceutics-14-01872-t001:** Main components of essential oils from *Citrus limon*, *Cupressus sempervirens*, *Litsea cubeba* and *Melaleuca alternifolia*, according to the supplier company.

Essential Oil	Part of the Plant	Extraction Method	Main Components
*C. limon*	Fruit	Cold pressing	Limonene (65.6%), β-pinene (15.06%), γ-terpinene (7.93%), α-pinene (2.34%), sabinene (1.76%) and myrcene (1.55%)
*C. sempervirens*	Leaf	NI	α-Pinene (52.4%), δ-3-carene (22%), limonene (3.5%), terpinolene (3.4%), myrcene (2.4%), terpenyl acetate (1.7%), cedrol (1.4%), β-pinene (1.2%) and terpinen-4-ol (1%)
*L. cubeba*	Fruit	Steam distillation	Geranyl acetate (42%), neral (30%) and limonene (13%)
*M. alternifolia*	Leaf	Steam distillation	Terpinen-4-ol (41%), γ-terpinene (20.5%), α-terpinene (9.63%), α-terpinolene (3.37%), α-terpineol (2.78%), α-pinene (2.59%), ρ-cymene (2.39%), aromadendrene (2%), vidiflorene (1.81%), δ-cadinene (1.54%) and 1,8-cineol (1.50%)

NI: not informed. Chemical analysis of the oils by chromatography was provided by the supplier company.

**Table 2 pharmaceutics-14-01872-t002:** Minimum inhibitory concentration (µg/mL) and minimum fungicidal concentration (µg/mL) of essential oils and clotrimazole tested with *Candida* species.

*Candida* spp. Isolates	*C. limon*		*C. sempervirens*		*L. cubeba*		*M. alternifolia*		Clotrimazole	
	MIC	MFC	MIC	MFC	MIC	MFC	MIC	MFC	MIC	MFC
*C. albicans* ATCC 90028	**2000**	1000	**2000**	4000	**1000**	1000	**2000**	4000	**0.25**	0.25
*C. albicans* SV 01	4000	>4000	4000	>4000	**2000**	4000	4000	>4000	**0.125**	0.25
*C. glabrata* ATCC 2001	**4000**	4000	**2000**	4000	**2000**	4000	**4000**	4000	**0.015**	0.030
*C. glabrata* SV 02	4000	>4000	**1000**	4000	**2000**	4000	4000	>4000	**0.25**	0.5
*C. krusei* ATCC 6258	4000	>4000	**2000**	4000	**1000**	2000	**4000**	4000	**0.5**	1
*C. krusei* SV 03	**1000**	4000	**500**	1000	**500**	1000	**2000**	4000	**0.5**	1
*C. parapsilosis* ATCC 22019	**1000**	4000	**4000**	4000	**1000**	4000	**4000**	4000	**0.25**	0.5
*C. parapsilosis* SV 04	4000	>4000	>4000	>4000	**2000**	4000	4000	>4000	**0.125**	0.25

MIC: minimum inhibitory concentration (µg/mL); MFC: minimum fungicidal concentration (µg/mL). Fungicide: (MFC/MIC: <4) in **bold**.

**Table 3 pharmaceutics-14-01872-t003:** Activity of essential oils and clotrimazole against the formation of biofilms and preformed biofilms of *Candida* species.

Species	*C. limon*			*C. sempervirens*			*L. cubeba*			*M. alternifolia*			*Clotrimazole*		
	MIC	MBIC	MBEC	MIC	MBIC	MBEC	MIC	MBIC	MBEC	MIC	MBIC	MBEC	MIC	MBIC	MBEC
*C. albicans* ATCC 90028	2000	4000	4000	2000	4000	>4000	**1000**	**1000**	**1000**	2000	2000	4000	0.25	1	1
*C. albicans* SV 01	4000	>4000	>4000	4000	4000	4000	2000	2000	4000	4000	4000	4000	0.125	0.5	1
*C. glabrata* ATCC 2001	4000	>4000	>4000	2000	>4000	4000	2000	2000	>4000	4000	4000	>4000	0.015	0.125	0.25
*C. glabrata* SV 02	4000	>4000	>4000	1000	>4000	>4000	2000	>4000	>4000	4000	4000	>4000	0.25	0.5	1
*C. krusei* ATCC 6258	4000	>4000	>4000	2000	2000	2000	**1000**	>4000	>4000	2000	4000	>4000	0.5	1	2
*C. krusei* SV 03	**1000**	2000	4000	**500**	**1000**	>4000	**500**	4000	>4000	**1000**	**1000**	>4000	0.5	1	1
*C. parapsilosis* ATCC 22019	**1000**	>4000	>4000	4000	4000	>4000	**1000**	4000	>4000	4000	4000	>4000	0.25	2	4
*C. parapsilosis* SV 04	4000	>4000	>4000	>4000	4000	>4000	2000	>4000	>4000	4000	>4000	>4000	0.125	0.25	0.5

Isolated MBIC and MBEC (µg/mL): capable of reducing ≥ 90% of optical density (OD) compared to control free of EO and clotrimazole. Results in **bold**: ≤1000 µg/mL.

**Table 4 pharmaceutics-14-01872-t004:** Minimum inhibitory, biofilm-inhibitory and biofilm-eradication concentrations of EO–EO and EO–clotrimazole combinations against *Candida* species.

Species	Combination	MIC (µg/mL)		Biofilm (µg/mL)			
		Isolated MIC *	Combined MIC **	Isolated MBIC *	Combined MBIC **	Isolated MBEC *	Combined MBEC **
*C. albicans* ATCC 90028	*M. alternifolia*	2000	250	2000	62.5	4000	62.5
	Clotrimazole	0.25	0.063	1	0.25	1	0.25
*C. albicans* SV 01	*L. cubeba*	2000	250	2000	125	4000	250
	*M. alternifolia*	4000	1000	4000	2000	4000	1000
*C. glabrata* ATCC 2001	*L. cubeba*	2000	500	2000	2000	>4000	2000
	*C. limon*	4000	1000	>4000	250	>4000	250
*C. glabrata* SV 02	*L. cubeba*	2000	250	>4000	1000	>4000	>1000
	*M. alternifolia*	4000	1000	4000	250	>4000	>250
	*C. limon*	4000	1000	>4000	4000	>4000	4000
	*M. alternifolia*	4000	1000	4000	250	>4000	250
*C. krusei* ATCC 6258	*C. sempervirens*	2000	1000	2000	4000	2000	>4000
	*C. limon*	4000	250	>4000	62.5	>4000	>250
	*C. limon*	1000	1000	>4000	4000	>4000	2000
	*M. alternifolia*	2000	1000	4000	250	>4000	500
*C. parapsilosis* SV 04	*C. sempervirens*	>4000	250	4000	500	>4000	125
	Clotrimazole	0.125	0.032	0.25	0.015	0.5	0.063

Isolated MBIC and MBEC (µg/mL): able to reduce by ≥90% optical density (OD) compared to control free of EO and clotrimazole. * Isolated: only one substance (EO or clotrimazole). ** Combined: MIC of the combination (EO–EO or EO–clotrimazole) that resulted in synergism.

**Table 5 pharmaceutics-14-01872-t005:** Mean survival rate (%) of *C. elegans* tested at different concentrations of essential oils from *C. limon*, *C. sempervirens*, *L. cubeba* and *M. alternifolia*.

Concentration (µg/mL)	*C. limon*	*C. sempervirens*	*L. cubeba*	*M. alternifolia*
250	95.83	97.83	97.83	97.82
500	85.96	95.35	87.76	94.00
1000	64.58	93.48	67.50	85.11
2000	58.82	92.86	51.11 *	46.15 *
4000	37.25 *	88.64	40.43	0

* LC_50_: lethal concentration responsible for the mortality of 50% of *C. elegans* larvae.

**Table 6 pharmaceutics-14-01872-t006:** Survival rate (%) of *C. elegans* subjected to combinations of essential oil and clotrimazole after 24 h of exposure.

Compound “A”	Concentration (µg/mL)	Compound “B”	Concentration (µg/mL)	Survival (% Average)
*M. alternifolia*	250	Clotrimazole	0.063	88.89
*L. cubeba*	250	*M. alternifolia*	1000	93.3
*C. sempervirens*	250	Clotrimazole	0.032	100.00
*C. sempervirens*	1000	*C. limon*	250	90.00
*C. limon*	1000	*M. alternifolia*	1000	18.52
*L. cubeba*	500	*C. limon*	1000	13.79

## Data Availability

Not applicable.

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
