# Peer review of "Combining Essential Oils with Each Other and with Clotrimazole Prevents the Formation of Candida Biofilms and Eradicates Mature Biofilms"

_pharmaceutics, 2022, doi:10.3390/pharmaceutics14091872_

Round 1

Reviewer 1 Report

Brief Summary

The manuscript pharmaceutics-1810897 analyzed the Citrus limonCupressus sempervirensLitsea cubeba and Melaleuca alternifolia essential oils and clotrimazole effects on Candida spp. biofilms. Essential oils, clotrimazole, and their combinations were also investigated for their toxicity, evaluating the survival of Caenorhabditis elegans after exposure to these agents. The study is interesting and could add knowledge to the field. The experiment was conducted with a good design and with valid methodologies. The data handling is suitable, and the quality of the manuscript preparation is appropriate. I would improve some aspects. See specific comments below. 

Introduction: The Introduction correctly places the study in a board context with a clear statement of the purpose of the research.

·         The working hypotheses being tested should be provided.

·         Some new references could be provided.

Materials and Methods: The authors described with sufficient detail the methods used.

·         Provide more details on clinical isolates (collection, isolation, accessions if available) or a valid reference.

Results: The results description is clear. To improve the table readability, I would change the less important values with some symbols, explaining it in the footnotes.

Discussion: The authors correctly discussed the results from the perspective of previous studies.

Conclusions: The section is appropriate and in line with the findings obtained. The authors should mention the importance of the findings obtained and some future studies.

Author Response

We are grateful for your contribution to the review of this manuscript. We agree with most of the reviewers' comments. The manuscript has been revised, and changes were made in the text and are highlighted in green. Our answers are punctuated in detail, according to reviewers.

Reviewer 1

  • Introduction: The Introduction correctly places the study in a board context with a clear statement of the purpose of the research. The working hypotheses being tested should be provided. Some new references could be provided.

The correction was done according to suggested. We have updated the references [1, 4, 5, 6, 9, 12, 13, 14, 15, 16] that address the topic described in the introduction.

  • Materials and Methods: The authors described with sufficient detail the methods used. Provide more details on clinical isolates (collection, isolation, accessions if available) or a valid reference.

The correction was done according to suggested. We refer to the article [23] that details the isolation of strains of clinical origin (L.:94-96).

  • Results:The results description is clear. To improve the table readability, I would change the less important values with some symbols, explaining it in the footnotes.

The correction was done according to suggested. To improve the view of concentrations, we have highlighted the relevant values in bold (Table 3).

  • Conclusions: The section is appropriate and in line with the findings obtained. The authors should mention the importance of the findings obtained and some future studies.

The correction was done according to suggested. There was description as a preliminary study for the development of formulations and topical pharmaceutical applications against Candida spp., described in the last paragraph of the conclusion, and focus of future studies (L.: 396-400).

The authors,   
Rafael Alves da Silva            
Denise Von Dolinger de Brito Röder
Reginaldo dos Santos Pedroso

Reviewer 2 Report

Accept in present form!

Author Response

We are grateful for your contribution to the review of this manuscript. We agree with most of the reviewers' comments. The manuscript has been revised, and changes were made in the text and are highlighted in green. Our answers are punctuated in detail, according to reviewers.

  • There are some grammatical errors in the text.

The review was carried out by native professionals who are experts in technical English in the field.

The authors,   
Rafael Alves da Silva            
Denise Von Dolinger de Brito Röder
Reginaldo dos Santos Pedroso

Reviewer 3 Report

This paper titled “Combining essential oils with each other and with clotrimazole prevents the formation of Candida biofilms and eradicates mature biofilms” is interesting. This manuscript could be considered for publication in Pharmaceutics after major revising. 

My comments are as follows:

  1. There are some grammatical errors in the text.
  2. Check the abbreviations and the text format of the full text. 
  3. The introduction section needs to be improved.
  4. In the discussion section, the authors should compare the current results with previous studies and highlight the advantage of this study.

Author Response

We are grateful for your contribution to the review of this manuscript. We agree with most of the reviewers' comments. The manuscript has been revised, and changes were made in the text and are highlighted in green. Our answers are punctuated in detail, according to reviewers.

  • There are some grammatical errors in the text.

The review was carried out by native professionals who are experts in technical English in the field.

  • Check the abbreviations and the text format of the full text. 

We have checked the abbreviations throughout the text carefully.

  • The introduction section needs to be improved.

 The correction was done according to suggested, introducing the guiding hypothesis of the research (L.:79-82).

  • In the discussion section, the authors should compare the current results with previous studies and highlight the advantage of this study.

There was description as a preliminary study for the development of formulations and topical pharmaceutical applications against Candida spp., described in the last paragraph of the conclusion, and focus of future studies (L.: 396-400).

The authors,   
Rafael Alves da Silva            
Denise Von Dolinger de Brito Röder
Reginaldo dos Santos Pedroso

Reviewer 4 Report

The manuscript Combining essential oils with each other and with clotrimazole prevents the formation of Candida biofilms and eradicates mature biofilms presents an approach of examination of the possible synergism between selected essential oils among themselves, and with clotrimazole.

Please consider making several changes:

L. 26 Synergism was performed by the checkerboard method – please rephrase the sentence. Consider English editing for the manuscript.

Table 3. Redesign the table in order to be more comprehensible.  

L.242. 3.3. Evaluation of synergism of essential oils and clotrimazole If you made a decision to present only synergisms, you should make other results available to the reviewers. Please add the rest of the data as supplementary material in your submission

 L.339. The evaluation of the toxicity of candidate products for new drugs has been carried out using alternative animal models and in human cells cultured in vitro, as a preliminary step to in vivo and clinical studies

The toxicity study is mandatory for every new drug candidate and there is no reason to cite 10 references. This is a well-known fact, and all studies have these steps. Please remove the references and rephrase the introductory sentence in this paragraph.

L.346-358 Please rewrite this paragraph in order to present your findings in logical order that is easy to follow. Comment one EO at a time. 

Author Response

We are grateful for your contribution to the review of this manuscript. We agree with most of the reviewers' comments. The manuscript has been revised, and changes were made in the text and are highlighted in green. Our answers are punctuated in detail, according to reviewers.

  • Synergism was performed by the checkerboard method – please rephrase the sentence. Consider English editing for the manuscript.

The correction was done according to suggested.  We rewrote the sentence as noted in L.26.

  • Table 3. Redesign the table in order to be more comprehensible.

We consider that the data arranged in this way in the table facilitate the observation and comparison of the data in the presented arrangement. The results are detailed enough for the reader to have a holistic view of the information. To improve the view of concentrations, we have highlighted the relevant values in bold.

  • 242. 3.3. Evaluation of synergism of essential oils and clotrimazole – If you made a decision to present only synergisms, you should make other results available to the reviewers. Please add the rest of the data as supplementary material in your submission.

We insert the data of the combinations performed as supplementary material, which is attached.

  • 339. The evaluation of the toxicity of candidate products for new drugs has been carried out using alternative animal models and in human cells cultured in vitro, as a preliminary step to in vivo and clinical studies. The toxicity study is mandatory for every new drug candidate and there is no reason to cite 10 references. This is a well-known fact, and all studies have these steps. Please remove the references and rephrase the introductory sentence in this paragraph.

The correction was done according to suggested (L.: 340-341).

  • 346-358 Please rewrite this paragraph in order to present your findings in logical order that is easy to follow. Comment one EO at a time. 

The correction was done according to suggested (L.: 347- 362).

The authors,   
Rafael Alves da Silva            
Denise Von Dolinger de Brito Röder
Reginaldo dos Santos Pedroso

Round 2

Reviewer 3 Report

Authors addressed all my comments